# Inhibition of Glutamine Uptake Resensitizes Paclitaxel Resistance in SKOV3-TR Ovarian Cancer Cell via mTORC1/S6K Signaling Pathway

**DOI:** 10.3390/ijms23158761

**Published:** 2022-08-06

**Authors:** Gyeongmi Kim, Se-Kyeong Jang, Yu Jin Kim, Hyeon-Ok Jin, Seunghee Bae, Jungil Hong, In-Chul Park, Jae Ho Lee

**Affiliations:** 1Division of Fusion Radiology Research, Korea Institute of Radiological & Medical Sciences, 75 Nowon-ro, Nowon-gu, Seoul 01812, Korea; 2Department of Cosmetics Engineering, Konkuk University, 120 Neungdong-ro, Gwangjin-gu, Seoul 05029, Korea; 3Department of Food and Microbial Technology, Seoul Women’s University, 621 Hwarangro, Nowon-gu, Seoul 01797, Korea; 4Department of Biological Engineering, Konkuk University, 120 Neungdong-ro, Gwangjin-gu, Seoul 05029, Korea; 5KIRAMS Radiation Biobank, Korea Institute of Radiological and Medical Sciences, 75 Nowon-ro, Nowon-gu, Seoul 01812, Korea

**Keywords:** ovarian cancer, chemoresistance, glutamine uptake, metabolic reprogramming, ASCT2, mTORC1, S6K

## Abstract

Ovarian cancer is a carcinoma that affects women and that has a high mortality rate. Overcoming paclitaxel resistance is important for clinical application. However, the effect of amino acid metabolism regulation on paclitaxel-resistant ovarian cancer is still unknown. In this study, the effect of an amino acid-deprived condition on paclitaxel resistance in paclitaxel-resistant SKOV3-TR cells was analyzed. We analyzed the cell viability of SKOV3-TR in culture conditions in which each of the 20 amino acids were deprived. As a result, the cell viability of the SKOV3-TR was significantly reduced in cultures deprived of arginine, glutamine, and lysine. Furthermore, we showed that the glutamine-deprived condition inhibited mTORC1/S6K signaling. The decreased cell viability and mTORC1/S6K signaling under glutamine-deprived conditions could be restored by glutamine and α-KG supplementation. Treatment with PF-4708671, a selective S6K inhibitor, and the selective glutamine transporter ASCT2 inhibitor V-9302 downregulated mTOR/S6K signaling and resensitized SKOV3-TR to paclitaxel. Immunoblotting showed the upregulation of Bcl-2 phosphorylation and a decrease in Mcl-1 expression in SKOV3-TR via the cotreatment of paclitaxel with PF-4708671 and V-9302. Collectively, this study demonstrates that the inhibition of glutamine uptake can resensitize SKOV3-TR to paclitaxel and represents a promising therapeutic target for overcoming paclitaxel resistance in ovarian cancer.

## 1. Introduction

Epithelial ovarian cancer remains the most lethal of all gynecologic malignancies [1]. At least 75% of patients with epithelial ovarian cancer are diagnosed with metastatic advanced disease (FIGO stage III and IV), which has high morbidity and mortality [2]. The combination of taxane and platinum is the current first-line chemotherapy regimen for advanced ovarian cancer [3]. Paclitaxel, a class of taxane drugs, has been identified as an effective drug in patients with recurrent ovarian cancer and is still being used. Paclitaxel binds to microtubules, stabilizes tubulin bundles, and stimulates the disassembly of microtubules in vivo [4]. As a result, cell proliferation is inhibited by halting the cell cycle at the metaphase/anaphase boundary by stabilizing the microtubule dynamics [5]. Although paclitaxel is the most effective anticancer agent for the treatment of advanced ovarian cancer to be used so far, there has been little development in therapeutic strategies or therapeutic adjuvants that could increase its efficacy. It has been reported that about 75% of patients with advanced ovarian cancer respond to the combination therapy of taxane and platinum [6]. However, the long-term survival rate of ovarian cancer patients is still poor due to recurrence and the development of resistance to first-line chemotherapeutic agents [7]. The emergence of chemoresistance has been a major hurdle in improving the survival rate of recurrent ovarian cancer [4]. Therefore, the identification of new therapeutic targets and the development of novel strategies to overcome chemoresistance are clinically important studies.

In cancers, proliferative capacity is sustained and allows cancer cells to grow constantly [8]. To satisfy this aberrant proliferation, cancer cells need large amounts of nutrients and utilize alternative metabolic processes [9,10]. Metabolic reprogramming is considered a core hallmark of cancer [10]. It has been reported that cancer cells commonly reprogram their metabolism to support cancer progression [11,12,13]. Cancer cells mainly require nutrient uptake and biosynthesis to carry out cell proliferation at an early stage, but as cancer progresses, metabolism is altered at later stages where metastasis and resistance are required. [14]. In addition to energy metabolism reprogramming, amino acid metabolism reprogramming also contributes to cancer progression [12,14]. For example, serine starvation inhibits cell growth in colorectal cancer cells in vitro and in vivo [15]. Restricting dietary serine and glycine can extend the survival time of tumor xenograft mice [16]. Several lines of evidence have indicated that tumor cells can control the expression of LAT-1 (SLC7A5), a major transporter of branched-chain amino acids (leucine, isoleucine, and valine) via oncogenic transcription factors such as c-Myc, HIF-2α, and Notch [17,18,19]. Argininosuccinate synthetase-1 (ASS-1), a rate-limiting enzyme in de novo arginine biosynthesis, has been reported to be downregulated in hepatocarcinoma and prostate cancer, as well as in malignant melanoma [20,21,22] and to be associated with advanced tumor stages and poor survival rates [22].

Glutamine is a nonessential amino acid, but many cancer cells depend on the uptake of extracellular glutamine for cancer progression [23]. Glutamine catabolism, also called glutaminolysis, begins to catalyze into glutamate by glutaminase (GLS) and is then converted into α-ketoglutarate (α-KG), which is used to supplement energy production in the TCA cycle [24]. Glutamine can also be used as a source for the biosynthesis of reduced glutathione and functions as a rate-limiting factor in cancer proliferation by providing an amide group in de novo nucleotide synthesis [24,25]. Several studies have shown that oncogenic mutations in cancer cells induce glutamine metabolism reprogramming [26,27,28]. In c-Myc-amplified cancer cells, glutamine deprivation selectively induces apoptosis [29]. Interestingly, it has been reported that c-Myc can enhance the transcriptional expression of glutamine transporters ASCT2 and SNAT5 (SLC38A5) [30,31]. The high glutamine requirement observed by reprogramming in cancer cells supports the importance of glutamine metabolism to promote the biosynthesis of building blocks along with the energy production required for cancer growth.

Mechanistic target of rapamycin (mTOR), a serine/threonine protein kinase that is ubiquitously expressed in mammalian cancer, is classified into two complexes, mTOR complex 1 (mTORC1) and mTOR complex 2 (mTORC2), and these complexes have distinct roles in cells [32]. Once the amino acids are transported into the cell, mTOR signaling is activated via different pathways depending on the kinds of amino acids [33]. The leucine that is sensed by sestrin2 interacts with the GATOR2 complex and activates mTORC1 [34]. Additionally, arginine induces the dissociation of CASTOR1, following the activation of mTORC1 signaling [35]. The main substrates for mTORC1 are ribosomal p70 S6 kinase (S6K) and eukaryotic translation initiation factor 4E-binding protein 1 (4E-BP1) [36]. S6K plays critical roles in protein synthesis, growth, and proliferation in cells [37], but it has been reported that S6K activity is frequently increased in cancer cells and that the elevated expression of S6K is associated with drug resistance in cancer [38].

Here, we report the characteristics of amino acid-deprived conditions in paclitaxel-resistant ovarian cancer SKOV3-TR cells. The effect of glutamine uptake modulation on cell proliferation and paclitaxel resistance in SKOV3-TR cells was evaluated. We also investigated the effects of the S6K selective inhibitor PF-4708671 and the glutamine transporter selective inhibitor V-9302, which mimic the regulation of amino acid metabolism on paclitaxel resistance in SKOV3-TR cells. These data suggest that modulating glutamine uptake has the potential to be a novel therapeutic target to overcome chemoresistance in ovarian cancer.

## 2. Results

### 2.1. The Cell Viabilities in Individual Amino Acid-Deprived Conditions in SKOV3-TR Cells

It has been reported that the excessive demand for nutrients in cancer cell growth is regulated by reprogrammed amino acid metabolism [10,39,40]. In this study, we tried to find a new anticancer target that regulates amino acid metabolism in paclitaxel-resistant ovarian cancer cells, which have limitations for the treatment of ovarian cancer. We first tested paclitaxel resistance in SKOV3-TR cells, which are paclitaxel-resistant cells, and compared the paclitaxel resistance of SKOV3 parental ovarian cancer cells. The SKOV3 and SKOV3-TR cells were treated with 1 to 100 nM of paclitaxel for 48 h. As shown in Figure 1A, paclitaxel caused a dose-dependent decrease in the viability of SKOV3 cells. However, paclitaxel did not affect the viability of SKOV3-TR cells. In addition, apoptotic cleaved PARP was increased in a dose-dependent manner in the paclitaxel-treated SKOV3 cell line, but not in the SKOV3-TR cell line (Figure 1B). We further confirmed the cytotoxic effect of SKOV3 and SKOV3-TR on paclitaxel via crystal violet staining for 48 h (Figure 1C). SKOV3 and SKOV3-TR cells were treated with 2.5 to 5000 nM of paclitaxel for 48 h. The IC50 values were defined as the drug concentration causing 50% growth inhibition compared to the controls. The IC50 values of paclitaxel in the SKOV3 and SKOV3-TR cells were obtained using dose-dependent curves at 48 h. The IC50 values on paclitaxel in SKOV3 and SKOV3-TR were measured to be 3.19 nM and 2176.01 nM. Since SKOV3-TR was highly resistant to paclitaxel, we used it to evaluate the cell cytotoxicity following incubation in each of the amino acid-deprived media. Interestingly, cell viabilities were significantly reduced to less than 50% in the arginine-, glutamine-, and lysine-deprived conditions in SKOV3-TR cells (Figure 1D). No significant differences were observed in the cell viability of parental SKOV3 cells with SKOV3-TR cells when cultured in each amino acid-deprived media (data not shown). Morphological observations collected via microscopy also confirmed the results above (Figure 1E). These results indicate that three amino acids—arginine, glutamine, and lysine—are essential for the survival of paclitaxel-resistant SKOV3-TR ovarian cancer cell line.

### 2.2. Glutamine-Deprived Culture Condition Inhibits mTORC1/S6K Signaling Pathway in SKOV3-TR Cells

Previous studies have determined that the mTORC1/S6K signaling pathway regulates the level of protein synthesis in cells under amino acid-deprived conditions in ovarian cancer cells [41]. As such, we investigated whether each of the amino acid-deprived culture conditions could modulate the mTORC1 signaling pathway in SKOV3-TR cells. Immunoblot analysis showed that the phosphorylation levels of S6K and S6 were downregulated in SKOV3-TR cells cultured with media deprived of arginine, glutamine, and lysine, but no changes were observed under glycine-deprived conditions (Figure 2A). Of the 20 amino acids, it has been determined that cancer cells require large amounts of glutamine to support their abnormally elevated growth rate [42]. Therefore, we focused on glutamine and investigated whether glutamine-deprived conditions affect cell viability and paclitaxel resistance in SKOV3-TR cells. Figure 2B showed that the glutamine-deprived culture condition significantly inhibited the phosphorylation of the mTOR, S6K, and S6 proteins within 12 h. We further investigated whether the regulation of mTORC1/S6K signaling by glutamine deprivation affected the proliferation of SKOV3-TR cells via glutamine supplementation. SKOV3-TR cells were cultured in glutamine-deprived medium for 12 h, and extra glutamine was added back. Then, the cell viability and the mTORC1/S6K signaling were analyzed. The results in Figure 2C show that the decreased cell viability caused by glutamine deprivation was gradually recovered by glutamine supplementation. Additionally, the phosphorylation levels of mTOR, S6K, and S6 proteins were recovered in a time-dependent manner. The phosphorylation of S6K was significantly restored 8 h after glutamine was added back to the medium (Figure 2D).

It has been reported that glutamine is metabolized to α-KG through glutaminolysis in cells, and that it directly affects the lysosomal translocation and activation of mTORC1 through GTP loading of RagB [42]. Therefore, we further examined whether SKOV3-TR could restore the phosphorylation of S6K and cell viability in glutamine-deprived culture conditions via supplementation with α-KG. MTT analysis showed that cell viability was restored in the glutamine-deprived culture when α-KG was supplemented (Figure 2E). The decreased phosphorylation of mTOR, S6K, and S6 levels was also increased via the supplementation of αKG in SKOV3-TR cells (Figure 2F). These data revealed that glutamine metabolism is a possible target for regulating cell survival through the modulation of mTORC1/S6K signaling in ovarian cancer cells.

### 2.3. Inhibition of S6K Resensitizes SKOV3-TR Cells to Paclitaxel

Since previous studies reported that activated mTOR signaling is a possible factor for drug resistance in several cancer cells and that regulating mTORC1/S6K signaling could be a potential strategy for cancer therapy [43,44], we hypothesized that S6K activity is related to paclitaxel resistance in SKOV3-TR cells. We checked whether the inhibition of mTORC1/S6K signaling through a specific S6K inhibitor, PF-4708671, could affect paclitaxel resistance in SKOV3-TR cells. Surprisingly, treatment with 20 μM of PF-4708671 reduced the phosphorylation of the S6 protein (Figure 3A) and significantly reduced cell viability upon cotreatment with paclitaxel (Figure 3B). In addition, to investigate the effects of S6K1 on paclitaxel resistance in SKOV3-TR, cell viability was examined in S6K1 knockdown SKOV3-TR using RNA interference. SKOV3-TR cells with S6K1 knockdown showed reduced cell viability caused by paclitaxel treatment compared to cells transfected with control siRNA (Figure 3C,D). These results suggest that the inhibition of S6K may be a strategy for resensitizing SKOV3-TR cells to paclitaxel and overcoming paclitaxel resistance in ovarian cancer.

### 2.4. Inhibition of the Glutamine Transporter ASCT2 Suppresses mTORC1/S6K Signaling and Resensitizes SKOV3-TR Cells to Paclitaxel

It has been reported that amino acid metabolism is related to drug resistance in cancer cells, as it regulates the TCA cycle and oxidative phosphorylation [45]. Additionally, several reports have indicated that inhibiting mTOR signaling can increase the sensitivity to paclitaxel [46,47,48]. Based on these studies, we investigated whether the regulation of glutamine metabolism could affect mTOR signaling and the paclitaxel resistance in the SKOV3-TR cell line. We used V-9302, a selective inhibitor of glutamine transporter ASCT2, to examine the effects of a glutamine-inhibited environment on SKOV3-TR cells. As shown in Figure 4A, the phosphorylation levels of the mTOR, S6K, and S6 proteins were reduced by treatment with 20 μM of V-9302 in SKOV3-TR cells, indicating that V-9302 effectively inhibited the mTORC1/S6K signaling pathway. MTT analysis showed that treatment with V-9302 marginally reduced the cell viability in SKOV3-TR cells, but interestingly, cotreatment with paclitaxel showed a higher inhibitory effect in cell viability (Figure 4B). To determine whether the resensitization effect of paclitaxel via V-9302 treatment was due to a direct effect on the ASCT2 protein, ASCT2 expression was knocked down in SKOV3-TR cells using a specific siRNA treatment, and the effect of paclitaxel on cell viability and mTORC1/S6K activity were then confirmed. As a result, mTORC1/S6K signaling was decreased by the siRNA-mediated knockdown of ASCT2 expression (Figure 4C), and cell viability was decreased by paclitaxel even more (Figure 4D). Our data indicate that the inhibition of glutamine uptake can effectively resensitize SKOV3-TR cells to paclitaxel.

### 2.5. Cotreatment Paclitaxel with S6K Inhibitor and Glutamine Uptake Inhibitor Induced the Phosphorylation of Bcl-2 and Decreased the Expression of Mcl-1 in SKOV3-TR Cells

Previous studies have identified that paclitaxel-induced apoptosis is mediated through various Bcl-2 families, including Bcl-2, Bax, and Bak [49,50,51], and in mitotic arrest-induced models, phosphorylation of Bcl-2 inhibits the antiapoptotic function of Bcl-2 [52]. We examined which Bcl-2 family mediated the apoptosis signaling pathway and restored paclitaxel sensitivity in glutamine metabolism-regulated SKOV3-TR cells. According to the results, cotreatment with V-9302 and paclitaxel markedly increased the phosphorylation of Bcl-2 at the serine 70 residue and downregulated the expression of Mcl-1 among the Bcl-2 family members in SKOV3-TR cells (Figure 5A). In addition, it was also confirmed that Bcl-2 phosphorylation was induced and that Mcl-1 expression was suppressed by treatment with 20 μM of PF-4708671 with paclitaxel in SKOV3-TR cells (Figure 5B). Furthermore, RT-PCR showed that Mcl-1 transcription was not reduced by cotreatment with either V-9302 or PF-4708671 with paclitaxel (Figure 5C), indicating that downregulating the Mcl-1 protein level was not due to transcriptional regulation. These data suggest that the inhibition of glutamine uptake and the downregulation of S6K activity can suppress paclitaxel resistance in SKOV3-TR cells, which may be associated with the phosphorylation of Bcl-2 at the serine 70 residue and the decrease in the Mcl-1 level.

## 3. Discussion

The results of our study demonstrate that inhibiting glutamine uptake resensitized paclitaxel resistance in SKOV3-TR cells via the downregulation of the mTORC1/S6K signaling pathway and that it is related to increased Bcl-2 phosphorylation and decreased Mcl-1 levels (Figure 6). These findings are expected to provide information on novel target therapies that inhibition of glutamine uptake can be applied to ovarian cancer and represent possibilities for overcoming paclitaxel resistance in ovarian cancer.

All nutrients are essential for growth and survival, and cancer cells in particular have been reported to provide building blocks using their reprogrammed metabolism for proliferation [8,10]. Although it has been reported that amino acid metabolism generally affects the mTOR signaling pathway and proliferation in various carcinomas [32,33,53], the possibility of the presence of cell-specific or cancer-specific amino acid metabolism has not been ruled out. As such, we examined the roles of each of the 20 amino acids in cancer survival and paclitaxel resistance in SKOV3-TR cells. Figure 1D showed that arginine, glutamine, and lysine deprivation play an important role in cell survival in SKOV3-TR cells. Additionally, we confirmed the effect of each of the 20 amino acids on cell survival in SKOV3 parental cells. Similar to the experimental results of SKOV3-TR, it was confirmed that cell viability was significantly inhibited in the arginine-, glutamine-, and lysine-deprived culture conditions in SKOV3 cells (data not shown), showing the potential of targeting amino acid metabolism to suppress survival in ovarian cancer, independent of paclitaxel resistance. Previous studies revealed that arginine deprivation induced cell cytotoxicity via the autophagy pathway in ovarian cancer cells [41], and that lysine deprivation enhanced the anticancer effect through L-lysine α-oxidase-induced oxidative stress [54]. Although glutamine deprivation-induced cytotoxicity has been reported, little is known about the effects of glutamine deprivation on paclitaxel resistance in ovarian cancer. Therefore, we focused on the function of glutamine metabolism to paclitaxel resistance in SKOV3-TR ovarian cancer cells.

It has been reported that glutamine not only serves as a critical component of protein synthesis in cells, but that it is also used to make nucleic acids and to generate energy in mitochondria [10,55]. Specifically, glutamine is enzymatically converted into α-KG and plays an important role in efficiently operating the tricarboxylic acid (TCA) cycle in mitochondria [55]. Recently, it has been reported that glutamine supplementation is sufficient to restore mTOR signaling during amino acid starvation, and α-KG also restores mTOR signaling in glutamine-deprived conditions [56,57]. In our results, mTOR signaling was restored by α-KG supplementation in SKOV3-TR cells under glutamine-deprived conditions (Figure 2F). However, the cell viability did not increase significantly (Figure 2E). This may be due to a limitation of α-KG supplementation in providing functions of glutamine other than energy metabolism.

In our study, we used V-9302, the selective ASCT2 glutamine transporter inhibitor, to mimic glutamine-deprived conditions. Treatment with V-9302 induced mTOR signaling downregulation that was similar to that observed in glutamine-deprived conditions, but the effects on cell viability were less than those observed in glutamine-deprived conditions (Figure 4B). Schulte et al. reported that the effect of V-9302 on cell viability varied across cancer cell lines [58]. This may be due to the reprogramming of glutamine metabolism in each cancer cell lines [9,10], or it may be the result of the factors involved in intracellular uptake other than ASCT2 [59,60]. Nevertheless, our experimental results indicated that treatment with V-9302 efficiently decreased mTORC1/S6K signaling and could affect paclitaxel resistance in SKOV3-TR (Figure 4A,B).

It has been reported that the Bcl-2 family is involved in the paclitaxel-induced apoptosis pathway in various cancer cells. Miller et al. showed paclitaxel-induced apoptosis via a Bak-dependent mechanism in breast cancer [61]. Srivastava et al. reported that Bcl-2 inhibits the phosphorylation of Bcl-2 and its antiapoptotic activity in ovarian cancer [62]. Bcl-2 phosphorylation works differently depending on the cellular state [52], but in mitotic arrest models, such as those of paclitaxel treatment, it has been reported that the Bcl-2 phosphorylation at the serine 70 residue could hinder heterodimerization with Bax, Bak, or other pro-apoptotic proteins, reducing the anti-apoptotic effect [63,64,65]. Additionally, Shuang et al. indicated that paclitaxel resistance in gastric cancer is associated with Mcl-1 stabilization [66]. Several reports have also indicated that the mTOR/S6K signaling may affect various Bcl-2 family proteins in various cells. It has been reported that the inhibition of S6K1 enhances the cytotoxicity of breast cancer cells via Bcl-2/ Bcl-xL inhibition [67]. Harada et al. have shown that S6K downregulates insulin-like growth factor-1 (IGF-1)-mediated Bad activation [68]. Although a relationship between paclitaxel and the intracellular Bcl-2 family has been reported [69], in which the Bcl-2 family acts in ovarian cancers that have been resensitized to paclitaxel, glutamine metabolism inhibition has not yet been elucidated. Our data indicate that the upregulation of Bcl-2 phsophorylation at the serine 70 residue and the inhibition of Mcl-1 in the intracellular expression level are possible cell death mechanisms induced by paclitaxel treatment under inhibition of glutamine metabolism in SKOV3-TR cells (Figure 5A,B). It has been reported that the phosphorylation of Bcl-2 at the serine 70 residue is critical in drug-induced apoptosis [70]. In addition, several reports have reported that Mcl-1 is not regulated at the transcriptional level, but that it is rapidly regulated in the intracellular expression level via the regulation of protein stability in the apoptosis pathway [71,72,73]. As shown in Figure 5C, no changes were observed in the Mcl-1 transcription levels after treatment with either V-9302 or PF-4708671 with paclitaxel. Therefore, the causes of cell death induced by cotreatment with V-9302 or PF-4708671 with paclitaxel may be an increase in the levels of phosphorylated Bcl-2 and a decrease in Mcl-1 in the cellular level.

Interestingly, it has been reported that mTORC2 stabilizes Mcl-1 via the inhibition of the glycogen synthase kinase 3 (GSK3)-dependent proteolytic degradation pathway [74]. We have not yet determined whether the inhibition of mTORC1/S6K signaling and cell death induced by cotreatment with either V-9302 or PF-4708671 with paclitaxel impair the protein stability of Mcl-1 directly or if they impair Mcl-1 indirectly through mTORC2 activation. Further research is needed to elucidate the detailed mechanism.

It has been reported that glutamine-deprived conditions and V-9302 can induce oxidative stress by mitochondrial dysfunction in various cells [75,76,77]. We did not exclude the possibility that paclitaxel resistance in glutamine-deprived conditions may be affected by the signaling mechanism induced by oxidative stress in SKOV3-TR cells, and we plan to analyze the possible mechanism of action through additional studies.

In this study, we examined the effect of glutamine uptake regulation on mTORC1/S6K signaling and the resensitization function of paclitaxel in SKOV3-TR cells. Although overcoming paclitaxel resistance through regulating glutamine metabolism may be an important point in cancer treatment, previous studies on the target of glutamine and the molecular mechanisms involved in glutamine absorption in humans are limited. Therefore, more studies focusing on clinical application in humans are still needed. Previous investigations have also demonstrated that cancer cells acquire a variety of cell- and tissue-specific metabolic processes by reprogrammed metabolism to adapt to environmental stress [9]. This study, which shows that treatment with V-9302 resensitized paclitaxel resistance in ovarian cancer, provides information regarding a novel anticancer target. V-9302 has already been studied to be applicable to various cancer cells, such as liver and triple-negative breast cancer [78,79]. It is considered to be more applicable to carcinomas that have acquired specific metabolic activity through reprogrammed metabolic processes.

In conclusion, this study provides us with a novel approach to overcome paclitaxel resistance in ovarian cancer through glutamine uptake inhibition and suggests the possibility of an application for anticancer drugs targeting various metabolic regulation processes.

## 4. Materials and Methods

### 4.1. Cell Culture and Reagents

The human ovarian cancer cell lines SKOV3 and paclitaxel-resistant SKOV3 (SKOV3-TR) were provided by Dr Anil K. Sood (The University of Texas MD Anderson Cancer Center, Houston, TX, USA) [80]. The cells were cultured in RPMI-1640 medium (#LM011-01; Welgene, Gyeongsan-si, Gyeongsangbuk-do, Republic of Korea) containing 10% fetal bovine serum (#S12450; R&D Systems, Minneapolis, MN, USA) in a humidified incubator at 37 °C and with 5% CO_2_. SKOV3-TR cells were cultured with 100 nM of paclitaxel (#10461; Cayman Chemical Company, Ann Arbor, MI, USA) to sustain their paclitaxel resistance. Each amino acid-deprived RPMI-1640 medium was generated by supplementing all amino acids (Arg, Asn, Asp, Cys, Glu, Gln, Gly, His, Hyp, Ile, Leu, Met, Phe, Pro, Ser, Thr, Trp, Tyr, Val; Welgene) at the concentrations contained in RPMI1640 (#LM011-01; Welgene), with the exception of the one in RPMI-1640 medium (#LM011-127; Welgene), which was supplemented with 10% dialyzed fetal bovine serum (#6400-044, Gibco; Thermo Fisher Scientific, Waltham, MA, USA). Thiazolyl blue tetrazolium bromide (MTT) was purchased from Sigma-Aldrich (#M5655, Merck KGaA, Darmstadt, Germany). α-ketoglutarate was purchased from Sigma-Aldrich (#K1128, Merck KGaA), V-9302 was purchased from MedChemExpress (#HY-112683; Monmouth Junction, NJ, USA), and PF-4708671 was purchased from Selleck Chemicals (#S2163; Houston, TX, USA)

### 4.2. Measurement of Cell Viability

Cell viability was measured using the MTT assay. Briefly, cells (1 × 10^5^) were seeded in 6-well plates and were further incubated for 18 h, and cells were treated with each drug for 48 h. After treatment, the cells were incubated with 1 mg/mL of MTT solution at 37 °C for 1 h. The supernatant was removed, and 2-propanol (#I915; Sigma-Aldrich; Merck KGaA) was added to dissolve formazan crystals. The absorbance was measured at the wavelength of 595 nm. The results were calculated as the percentage of cell viability relative to the untreated control, and each experiment was repeated 3 times.

### 4.3. Crystal Violet Staining

SKOV3 and SKOV3-TR cells (5 × 10^4^) were seeded in 12-well culture plates for 18 h and were then treated with 2.5 to 5000 nM of paclitaxel for 48 h. Cells were stained with 0.5% crystal violet (BIOPURE, Seoul, Republic of Korea) for 30 min. The stained cells were solubilized via the addition of methanol and were incubated at room temperature for 1 h. Subsequently, the absorbance of the solution was determined at 595 nm. IC50 values were estimated using the Quest Graph IC50 Calculator (AAT Bioquest, Inc., Sunnyvale, CA, USA) [81].

### 4.4. Transfection

ASCT2 (#sc-60210), S6K1 (#sc-36165), and control (#sc-37007) siRNAs were purchased from Santa Cruz Biotechnology (Dallas, TX, USA). Lipofectamine RNAiMAX transfection reagent (#13778; Invitrogen; Thermo Fisher Scientific) was used for siRNA transfection following the manufacturer’s protocol. For a 6-well plate, the final volume of siRNA (10 μmol/L) used per well was 10 μL (100 pmol), and the final volume of Lipofectamine RNAiMAX used per well was 9 μL. siRNA-lipofectamine RNAiMAX complexes were added to the cells for 6 h in serum-free medium, and the medium was replaced with fresh serum medium after transfection. Experiments were performed 24 h after transfection.

### 4.5. Immunoblot Analysis

Cells (1 × 10^5^) were seeded in 6-well plates and were further incubated for 18 h before drug treatment. The drug-treated cells were washed with cold PBS and lysed with cell lysis buffer (#9803; Cell Signaling Technology, Beverly, MA, USA) containing protease inhibitor cocktail (#G6521; Promega, Madison, WI, USA) and phosphatase inhibitor cocktail (#78420; Thermo Fisher Scientific). The protein concentration was determined using Bradford reagent (#5000006; Bio-Rad Laboratories, Hercules, CA, USA). Twenty micrograms of total protein was separated by sodium dodecyl sulfate-polyacrylamide gel electrophoresis and was transferred to nitrocellulose membranes. The membranes were incubated with specific primary antibodies overnight at 4 °C and horseradish peroxidase-conjugated secondary antibodies for 1 h at room temperature. The blots were detected using SuperSignal West Pico chemiluminescence substrates (Pierce; Thermo Fisher Scientific) and captured on X-ray film. The following antibodies were used: antibody against cleaved PARP (#9541), mTOR (#2983), p-mTOR (S2448) (#2971), S6K (#9202), p-S6K (Thr389) (#9205), S6 (#2217), p-S6 (Ser240/244) (#5364), p-Bcl-2 (#2827), Bcl-2 (#150071), Bcl-XL (#2764), Bax (#2772), p-Bad (#5284), Bad (#9292), Bid (#2002), Bak (#12105), and Mcl-1 (#5453), all of which were obtained from Cell Signaling Technology. The antibody against XIAP (#610763) was obtained from BD Biosciences, and the antibody against β-actin (#A5316) was obtained from Sigma-Aldrich (Merck KGaA). The intensities of the protein bands were quantified with ImageJ software (National Institutes of Health, Bethesda, MD, USA).

### 4.6. RNA Extraction and Reverse Transcription PCR Analysis

Cells (1 × 10^5^) were seeded in 6-well plates and were incubated overnight. Then, the cells were treated with each drug followed by incubation for 24 h. Total RNA was isolated from cells treated with each reagent using TRIzol reagent according to the manufacturer’s instructions (#15596-026, Invitrogen; Thermo Fisher Scientific). One microgram of total RNAs was reverse transcribed using M-MLV-Reverse Transcriptase (Invitrogen; Thermo Fisher Scientific) and the complementary DNAs were used for PCR template. The following primers were used for the amplification of specific genes: Mcl-1 (5′-CCA GTA CGG ACG GGT CAC TA-3′ and 5′-CCC CAG TTT GTT ACG CCG TC-3′) and β-Actin (5′-GGA TTC CTA TGT GGG CGA CAG-3′ and 5′-CGC TCG GTG AGG ATC TTC ATG-3′)

### 4.7. Statistical Analysis

The data are presented as the mean ± standard deviation (SD) of three independent experiments. Statistical differences were determined by one-way ANOVA followed by Turkey’s test using the GraphPad Prism software (version 8.0.1, San Diego, CA, USA). Differences of *p* < 0.05 were considered statistically significant. 

## Figures and Tables

**Figure 1 ijms-23-08761-f001:**
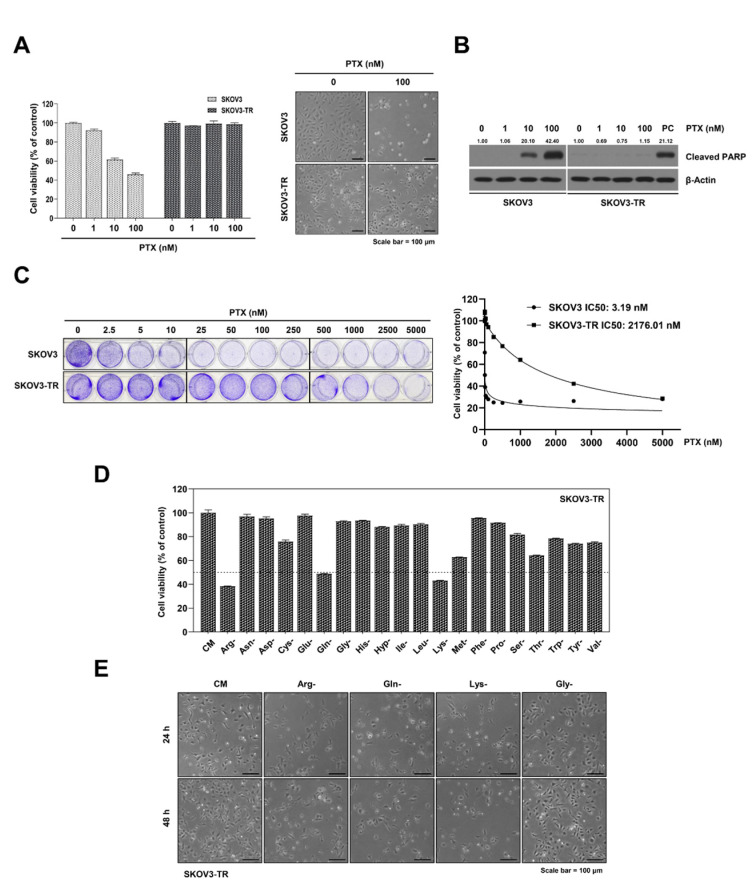
Differential effect on cell viability by individual amino acid-deprived conditions in SKOV3-TR cells. (**A**–**C**) SKOV3 and SKOV3-TR cells were cultured in medium treated with indicated concentrations of paclitaxel (PTX) for 48 h. (**A**) The cell viabilities of SKOV3 and SKOV3-TR cells were measured by MTT assay, and cells were visualized using a microscope after treatment with 100 nM of paclitaxel. (**B**) The apoptotic cleaved PARP was analyzed by immunoblotting, PC (positive control); SKOV3-TR cells treated with 2000 nM of paclitaxel. Protein expression was quantified as fold changes with respect to the control after normalization of respective β-actin levels using ImageJ software. (**C**) The cytotoxic effect of SKOV3 and SKOV3-TR cells on paclitaxel were measured by crystal violet staining. (**D**) SKOV3-TR cells were cultured in medium deprived of each of the 20 amino acids for 48 h. Cell viabilities were measured by MTT assay. (**E**) Microscopic observations of SKOV3-TR cells in arginine-, glutamine-, lysine- and glycine-deprived culture conditions after 48 h. The data are presented as the mean percentage of control. CM, complete media; PTX, paclitaxel.

**Figure 2 ijms-23-08761-f002:**
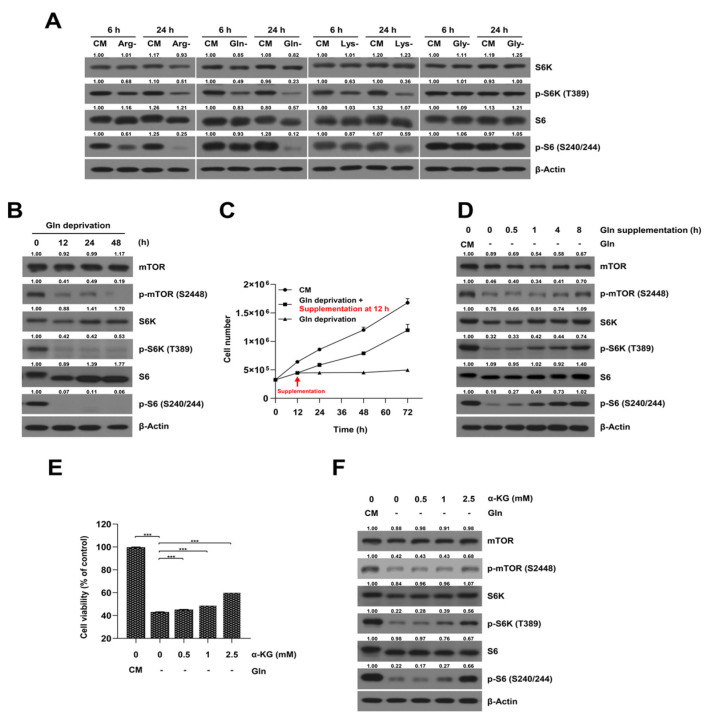
The effects of amino acid deprivation on mTORC1/S6K1 activity in SKOV3-TR cells. (**A**) SKOV3-TR cells were cultured in media deprived of each of the 20 amino acids. Using immunoblotting, the expression levels of S6K and S6 proteins and their phosphorylation were analyzed in culture conditions deprived of arginine, glutamine, lysine, and glycine. (**B**) SKOV3-TR cells were cultured in glutamine-deprived conditions for the indicated times. The expression levels of the mTOR, S6K, and S6 proteins were detected by immunoblotting. (**C**,**D**) SKOV3-TR cells were cultured in glutamine-deprived medium for 12 h and then supplemented with glutamine, and they were further incubated for the indicated times. (**C**) Live cells were counted using trypan blue staining. (**D**) The expression of mTOR, S6K, and S6 proteins and their phosphorylation were measured by immunoblotting. (**E**,**F**) SKOV3-TR cells were cultured in medium deprived of glutamine for 12 h and then supplemented with α-KG at the indicated doses for 48 h. (**E**) Cell viabilities were measured by MTT assay. (**F**) The expression of mTOR, S6K, and S6 proteins and their phosphorylation were analyzed by immunoblotting. Protein expression was quantified as fold changes with respect to the control after normalization of respective β-actin levels using ImageJ software. The data are presented as the mean percentage of control ± SD relative to the control (*n* = 3, *** *p* < 0.001). CM, complete media; PTX, paclitaxel; α-KG, α-ketoglutarate.

**Figure 3 ijms-23-08761-f003:**
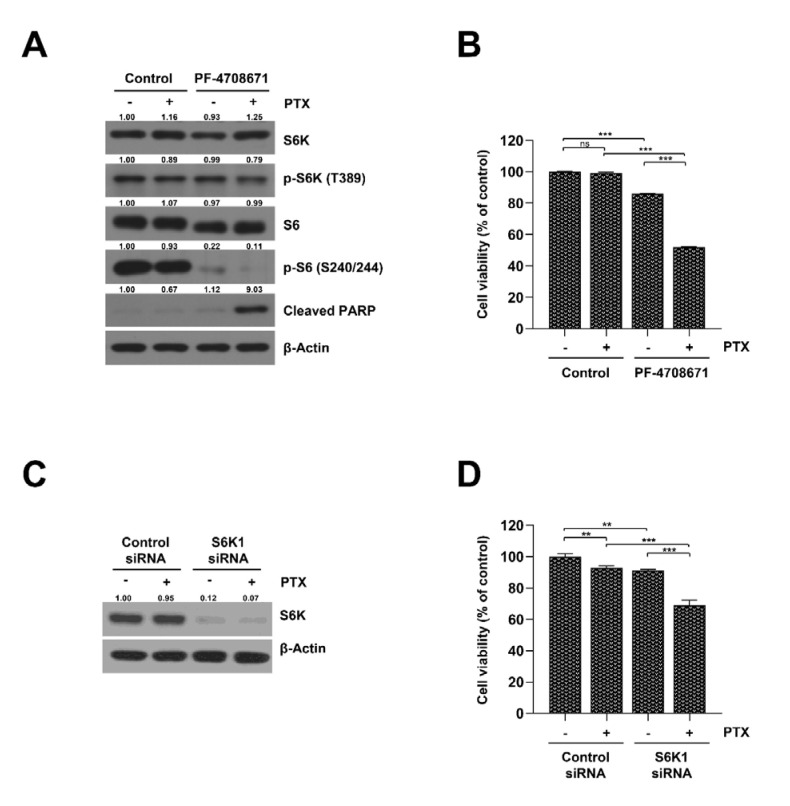
Inhibition of S6K restores the paclitaxel sensitivity in SKOV3-TR cells. (**A**,**B**) SKOV3-TR cells were cultured in medium treated with 100 nM of paclitaxel and 20 μM of PF-4708671 for 48 h. (**A**) The expression of S6K and S6 proteins and their phosphorylation were measured by immunoblotting. (**B**) The cell viabilities were measured by MTT assay. (**C**,**D**) SKOV3-TR cells were transfected with control and S6K1 siRNA for 6 h and were subsequently treated with 100 nM of paclitaxel for 48 h. (**C**) The expression of S6K and S6 proteins and their phosphorylation were measured by immunoblotting. (**D**) The cell viabilities were measured by MTT assay. Protein expression was quantified as fold changes with respect to the control after normalization of respective β-actin levels using ImageJ software. The data are presented as the mean percentage of control ± SD relative to the control (*n* = 3, ** *p* < 0.01; *** *p* < 0.001; ns, not significantly different). PTX, paclitaxel.

**Figure 4 ijms-23-08761-f004:**
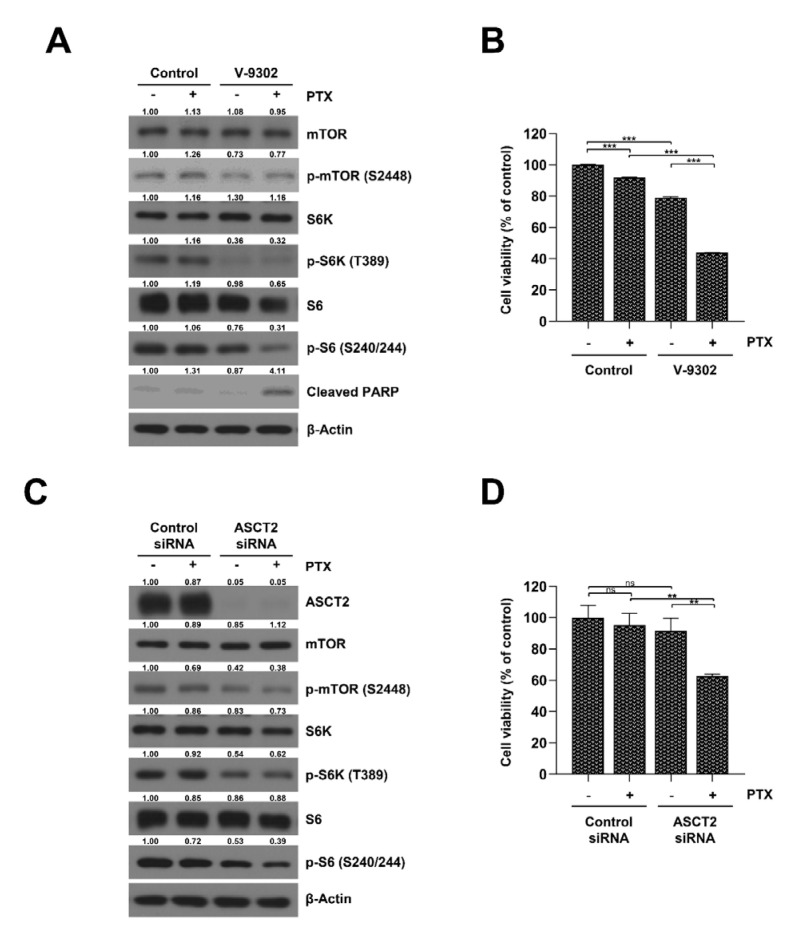
Inhibition of glutamine transporter ASCT2 suppresses mTORC1/S6K activity and resensitizes SKOV3-TR cells to paclitaxel. (**A**,**B**) SKOV3-TR cells were cultured in medium treated with 100 nM of paclitaxel and 20 μM of V-9302 for 48 h. (**A**) The expression of mTOR, S6K and S6 proteins and their phosphorylation were measured by immunoblotting. (**B**) The cell viabilities were measured by MTT assay. (**C**,**D**) SKOV3-TR cells were transiently transfected with control and ASCT2 siRNA for 6 h and were subsequently treated with 100 nM of paclitaxel for 48 h. (**C**) The expression of mTOR, S6K, and S6 proteins and their phosphorylation were measured by immunoblotting (**D**). The cell viabilities were measured by MTT assay. Protein expression was quantified as fold changes with respect to the control after normalization of respective β-actin levels using ImageJ software. The data are presented as the mean percentage of control ±SD relative to the control (*n* = 3, ** *p* < 0.01; *** *p* < 0.001; ns, not significantly different). PTX, paclitaxel.

**Figure 5 ijms-23-08761-f005:**
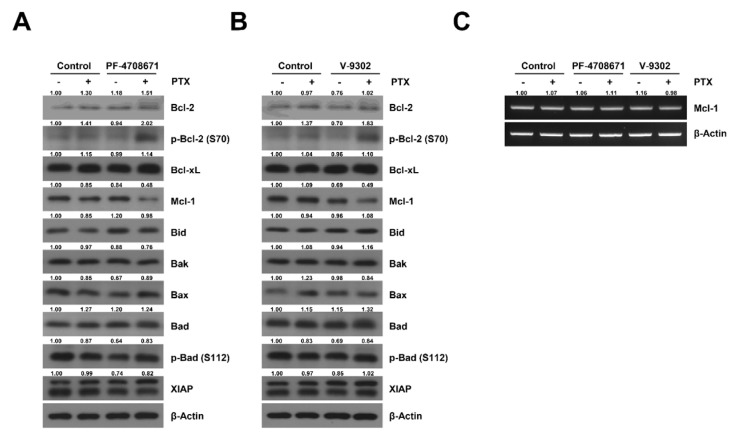
Cotreatment with PF-4708671 or V-9302 and paclitaxel upregulates the phosphorylation of Bcl-2 and downregulates the cellular level of Mcl-1 in SKOV3-TR cells. (**A**,**B**) SKOV3-TR cells were cotreated with 20 μM of PF-4708671 or 20 μM of V-9302 and 100 nM paclitaxel for 48 h, and the expression of the Bcl-2 family (Bcl-2, Bcl-xL, Mcl-1, Bid, Bak, Bax, and Bad), the phosphorylation of Bcl-2 at the serine 70 residue and Bad at serine 112 residue, and XIAP were then analyzed by immunoblotting. (**C**) Transcriptional level of Mcl-1 was analyzed by RT-PCR in SKOV3-TR cells cotreated with PF-4708671 or V-9302 and paclitaxel for 48 h. Protein expression was quantified as fold changes with respect to the control after normalization of respective β-actin levels using ImageJ software. PTX, paclitaxel.

**Figure 6 ijms-23-08761-f006:**
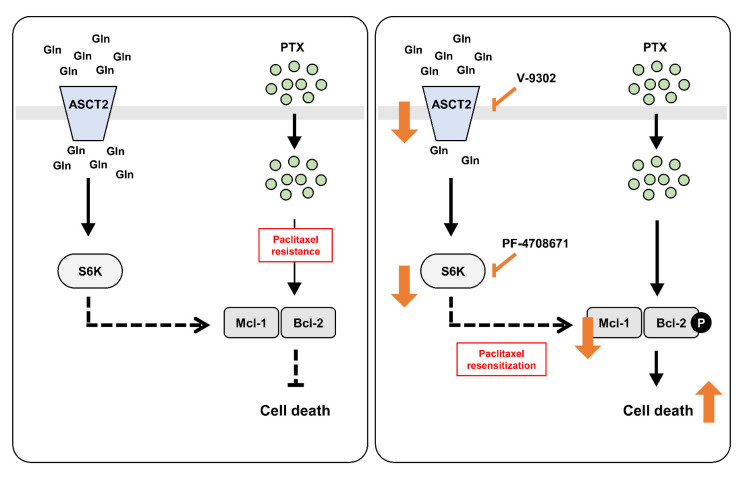
Schematic depiction of the resensitization mechanism for paclitaxel in SKOV3-TR via the inhibition of glutamine uptake and mTORC1/S6K signaling. In SKOV3-TR cells, inhibition of the glutamine transporter and mTORC1/S6K signaling by V-9302 and PF-4708671 resensitized the paclitaxel resistance, respectively, which regulated the Mcl-1 level and phosphorylation of Bcl-2. The cassettes represent cells, and the gray barrier is the cell membrane. PTX, paclitaxel.

## Data Availability

Not applicable.

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
