# Peer review of "Inhibition of Glutamine Uptake Resensitizes Paclitaxel Resistance in SKOV3-TR Ovarian Cancer Cell via mTORC1/S6K Signaling Pathway"

_ijms, 2022, doi:10.3390/ijms23158761_

Round 1
Reviewer 1 Report
See attachment

Author Response
(Comment 1) In Figure 1A the authors show cell viabilities of SKOV3 and SKOV3-TR cells. From the data it seems that the IC50 for SKOV3 cells is about 100nM and unknown for SKOV3-TR cells. The authors must show complete dose response curves (rather than box plots) for parental and SKOV3-TR cells and indicate IC50 values in it. To show the difference in their viabilities, the authors should also perform clonogenic assay to illustrate stark difference between drug sensitive and drug resistant line.
→ (Response) As reviewer’s suggestion, to investigate whether the difference in their viabilities of SKOV3 and SKOV3-TR, we examined crystal violet staining instead of clonogenic assay. IC50 values on paclitaxel in SKOV3 and SKOV3-TR were measured to be 3.19 nM and 2176.01 nM. In the revised manuscript, additional data are included as Figures 1C; Material and methods, Results and Figure legends are subsequently updated.
(Comment 2) Glutamine deprivation affecting cell viability and mTORC signaling are shown in Figure 2C but its effect on paclitaxel resistance should be shown.
→ (Response) In Figure 2, we examined the effect of glutamine depletion on cell viability and mTORC signaling in the paclitaxel-resistant ovarian cancer cells (SKOV3-TR). In view of the clinical relevance, glutamine depletion condition could not exist. Therefore, we examined effect of inhibition of ASCT2, a glutamine transporter, by siRNA and V9302 in SKOV3-TR cells. The results showed that the inhibition of glutamine uptake with ASCT2 siRNA and V-9302 enhanced the sensitivity to paclitaxel in paclitaxel-resistant cells (Figure 4). We suggest that activation of mTORC1 signaling by glutamine uptake is one of the reason inducing the resistance of paclitaxel in ovarian cancer.
(Comment 3) In several of the western blots that are shown in the manuscript, the images (few examples are shown below) that appear in the manuscript (on the left) looks drastically different from the original blots (on the right) in the supplementary data. This is unacceptable. It is clear that the authors have cropped, compressed and forced the images to fit into thinner looking bands as presented in the manuscript which must be seriously corrected. Moreover, the authors must quantify every western blots and provide plots of protein of interest with respective loading control normalized values to draw optimal conclusions from the result.
→ (Response) Thank you for your valuable comment. As reviewer’s suggestion, in the revised manuscript overall western blot images were shown without cropping and compression to fit the thinner bands. The protein expression was quantified using ImageJ software and fold change with respect to control after normalization to respective actin and indicated by numbers.
(Comment 4) The manuscript needs extensive revision for language and grammar. Even the symbols denoted in authors like * and # are not properly corresponded in the affiliation section. The authors must go through proper proofreading before next submission.
→ (Response) As reviewer’s suggestion, this manuscript has been English correct edited by MDPI’s editing service. Additionally, we have corrected the symbols marked with †, *.
Reviewer 2 Report
The manuscript is an original study, which presents a particular aspect of drug resistance in ovarian cancer. The authors display metabolomic targets of paclitaxel resistance and point towards potential therapeutic applications. All conclusions are supported by several well conducted experiments, and overall, it is a well written paper. I suggest accepting the paper, with some minor revisions:
Page 6, chapter 2.3 please indicate the concentration of PF-4708671 as well in row 199, not only in the figure caption. Same for V-9302 in chapter 2.4.
Page 9, Figure 6- it is a very suggestive figure, but please explain that the gray barrier is the cell membrane, the cassettes represent cells, and so on.
Page 11, chapter 4.1 the origin of the SKOV3 and SKOV3-TR cell lines should be stated. If SKOV3-TR was not purchased originally from a cell bank, but developed at the donor institution, a reference about the method is needed. Row 389 “by supplementing all amino acids” please indicate their concentration, for each one.
Chapter 4.3 More details about the methods should be inserted
Overall in the Methods chapter; the city/country of the provider should be inserted everywhere.
4.6- the following fragment have to be deleted: “ The Materials and Methods should be described [.........] while well-established methods can be briefly described and appropriately cited.”
Author Response
(Comment 1) Page 6, chapter 2.3 please indicate the concentration of PF-4708671 as well in row 199, not only in the figure caption. Same for V-9302 in chapter 2.4.
→ (Response) In the revised manuscript, the chapter 2.3 and 2.4 has been rewritten.
(Comment 2) Page 9, Figure 6- it is a very suggestive figure, but please explain that the gray barrier is the cell membrane, the cassettes represent cells, and so on.
→ (Response) In the revised manuscript, the figure 6 description has been rewritten.
(Comment 3) Page 11, chapter 4.1 the origin of the SKOV3 and SKOV3-TR cell lines should be stated. If SKOV3-TR was not purchased originally from a cell bank, but developed at the donor institution, a reference about the method is needed. Row 389 “by supplementing all amino acids” please indicate their concentration, for each one.
→ (Response) In the revised manuscript, the chapter 4.1 has been rewritten. SKOV3 and SKOV3-TR cells were gifted from Dr. Anil K. Sood when Dr. Tae Jin Kim and Jae Ho Lee were worked in Cheil General Women’s Healthcare Center.
(Comment 4) Chapter 4.3 More details about the methods should be inserted. Overall in the Methods chapter; the city/country of the provider should be inserted everywhere.
→ (Response) In the revised manuscript, the chapter 4.4(4.3) has been rewritten.
(Comment 5) 4.6- the following fragment have to be deleted: “The Materials and Methods should be described [.........] while well-established methods can be briefly described and appropriately cited.”
→ (Response) This sentence has been deleted.
Reviewer 3 Report
The manuscript is an original study, which presents a particular aspect of drug resistance in ovarian cancer. The authors display the metabolomic targets of paclitaxel resistance and point towards potential diagnostic and therapeutic applications. The conclusions of the paper are based on several well conducted experiments, and overall it is a well written paper. I suggest minor corrections, such:
Page 3, chapter 2.1 please reformulate "As shown in Fig 1A, MTT analysis and 119 microscopic observation showed the cell cytotoxicity in SKOV3 cells"
Page 7, chapter 2.4. the concentration of V- 9302 could be useful. Same for PF-4708671 in 2.5
Page 11, 4.1- the origin of SKOV3 and SKOV3-TR cell line should be mentioned (ATCC, ECACC, or others). If SKOV-TR is not commercially available, the authors should refer the method to obtain this cells line
The provider of each amino-acid is missing, as well as the concentration of aminoacids in the media.
4.3 some details about the transfection should be inserted
4.4 Reformulation needed: "Cells (1×105) were seeded in 6-well plate and further incubated for 18 h before drug 410 treatment." the interval of pre-therapy cultivation and the duration of the therapy must be clarified
4.6- the phrases: "The Materials and Methods should be described with sufficient details [....] New methods and protocols should be described in detail while well-established methods can be briefly described and appropriately cited." have to be be erased
References- authors should follow the journal's indications.
Author Response
(Comment 1) Page 3, chapter 2.1 please reformulate "As shown in Fig 1A, MTT analysis and 119 microscopic observation showed the cell cytotoxicity in SKOV3 cells"
→ (Response) This sentence has been rewritten as follows: “As shown in Figure 1A, paclitaxel caused a dose-dependent decrease in the viability of SKOV3 cells. However, paclitaxel did not affect the viability of SKOV3-TR cells.”
(Comment 2) Page 7, chapter 2.4. the concentration of V- 9302 could be useful. Same for PF-4708671 in 2.5
→ (Response) In the revised manuscript, the chapter 2.3 and 2.5 has been rewritten.
(Comment 3) Page 11, 4.1- the origin of SKOV3 and SKOV3-TR cell line should be mentioned (ATCC, ECACC, or others). If SKOV-TR is not commercially available, the authors should refer the method to obtain this cells line. The provider of each amino-acid is missing, as well as the concentration of aminoacids in the media.
→ (Response) In the revised manuscript, the chapter 4.1 has been rewritten.
(Comment 4) 4.3 some details about the transfection should be inserted
→ (Response) In the revised manuscript, the chapter 4.3 has been rewritten.
(Comment 5) 4.4 Reformulation needed: "Cells (1×105) were seeded in 6-well plate and further incubated for 18 h before drug 410 treatment." the interval of pre-therapy cultivation and the duration of the therapy must be clarified
→ (Response) This sentence has been rewritten as follows: “ Cells (1×) were seeded in 6-well plate and further incubated for 18 h and cells were treated each drug for 48 h.”
(Comment 6) 4.6- the phrases: "The Materials and Methods should be described with sufficient details [....] New methods and protocols should be described in detail while well-established methods can be briefly described and appropriately cited." have to be erased
→ (Response) This sentence has been erased.
(Comment 7) References- authors should follow the journal's indications.
→ (Response) In the revised manuscript, the references format has been updated following the journal’s indication.
Reviewer 4 Report
A well written paper bringing into attention teh important issue of ovarian cancer. Although is a highly chemo sensitive disease, the rate of relapse is high thansforming it into an incurable disease with a need for continuum of care. Paclitaxel is part of the chemo backbone for this disease, but the resistance to paclitaxel is aalso a challenge in ovarian cancer so any research designed to overcome resistance is very important.
As the authors stated their findings are expected to provide information on novel target therapies in the glutamine uptake that can be applied to ovarian cancer and possibilities for overcoming the paclitaxel resistance in ovarian cancer.
Because little is known about glutamine deprivation effect on paclitaxel resistance in ovarian cancer teh authors focused on the function of glutamine metabolism to paclitaxel resistance in SKOV3-TR ovarian cancer cells.
In conclusion the study adds new data and a novel approach to overcome paclitaxel resistance in ovarian cancer through glutamine uptake inhibition and suggests the possibility of development of new anticancer drugs targeting metabolic regulation.
Author Response
Thank you for your valuable comments.
Round 2
Reviewer 1 Report
The authors have addressed the issues raised during revision. I agree with the changes and recommend the manuscript to be published.